# Novel Synthesis Route of Plasmonic CuS Quantum Dots as Efficient Co-Catalysts to TiO_2_/Ti for Light-Assisted Water Splitting

**DOI:** 10.3390/nano14191581

**Published:** 2024-09-30

**Authors:** Larissa Chaperman, Samiha Chaguetmi, Bingbing Deng, Sarra Gam-Derrouich, Sophie Nowak, Fayna Mammeri, Souad Ammar

**Affiliations:** 1Université Paris Cité, CNRS UMR-7086, ITODYS, 75205 Paris, France; larissa.chaperman@univ-paris-diderot.fr (L.C.); bingbing.deng@univ-paris-diderot.fr (B.D.); sarra.derouich@u-paris.fr (S.G.-D.); sophie.nowak@u-paris.fr (S.N.); fayna.mammeri@u-paris.fr (F.M.); 2Faculté des Sciences, Université 20-Août-1955-Skikda, Skikda 21000, Algeria; s.chaguetmi@univ-skikda.dz

**Keywords:** plasmonic semiconductive nanoparticles, polyol process, titania photoanodes, water splitting

## Abstract

Self-doped CuS nanoparticles (NPs) were successfully synthesized via microwave-assisted polyol process to act as co-catalysts to TiO_2_ nanofiber (NF)-based photoanodes to achieve higher photocurrents on visible light-assisted water electrolysis. The strategy adopted to perform the copper cation sulfidation in polyol allowed us to overcome the challenges associated with the copper cation reactivity and particle size control. The impregnation of the CuS NPs on TiO_2_ NFs synthesized via hydrothermal corrosion of a metallic Ti support resulted in composites with increased visible and near-infrared light absorption compared to the pristine support. This allows an improved overall efficiency of water oxidation (and consequently hydrogen generation at the Pt counter electrode) in passive electrolyte (pH = 7) even at 0 V bias. These low-cost and easy-to-achieve composite materials represent a promising alternative to those involving highly toxic co-catalysts.

## 1. Introduction

The quest for sustainable and efficient energy solutions has led to significant interest in the development of advanced materials for water splitting applications. Among these, semiconductive nanostructures play a crucial role in the photoelectrochemical (PEC) water decomposition reactions, which harnesses solar energy to produce hydrogen fuel. TiO_2_ is among the most prolific materials for light-assisted water electrolysis thanks to its excellent chemical stability and strong photooxidative capabilities, with a band structure triggering water redox potentials [1,2,3]. Nevertheless, TiO_2_ presents significant challenges due to its wide bandgap (approximately 3.2 eV for anatase), which limits its light absorption to the UV region, a small fraction of the solar spectrum. Moreover, TiO_2_ suffers from the rapid recombination of photogenerated electron-hole pairs, which severely impacts its photocatalytic efficiency, limiting electron transfer from the TiO_2_ conduction band (CB) to the external circuit and then to the cathode materials to be scavenged for proton reduction and hydrogen generation.

A wide variety of strategies was developed to overcome these limitations, the elaboration of titania-based semiconductive hetero-nanostructures being the object of particularly intensive research [4,5]. In such studies, TiO_2_-based structures are combined to one (or more) sensitizer, a semiconductor with fine-tuned properties that allow to compensate for the TiO_2_ shortcomings and improve the overall efficiency of the resulting system. The chosen semiconductor must possess appropriate electronic band structures with small bandgap energy and a high absorption coefficient in order to harvest sufficient solar photons. In addition, the energetic levels of its band structure need to straddle the redox potential of the desired water redox reactions, satisfying both the thermodynamics and kinetics requirements for conducting efficient photocatalytic reactions. Furthermore, chemical robustness and photostability are also essential for the selected semiconductors to be considered for such applications to allow for long-term, stable activity in photocatalytic processes [4]. Finally, such materials must be of low toxicity to avoid any deleterious effects on human operators as well as any environmental contamination.

The photocatalytic efficiency of the resulting semiconductive hetero-nanostructures is highly dependent on both the intrinsic photon absorption capability and charge transfer dynamics of the two (or more) photoanode components. Light absorption and charge transfer capabilities are mostly inherent to the band gap of the selected materials and their doping, including self-doping. The narrower the band gap, the higher the photogenerated charge density. As the doping rate increases, the conductivity increases. So, in a standard photoelecrochemical (PEC) cell, in ideal conditions, upon illumination, the photogenerated hole carriers are transported from the bulk of the semiconductor to its electrolyte interface, where they may participate in a water oxidation reaction. Conversely, the photogenerated electron carriers are transported from the bulk of the semiconductor to its titanium interface to be collected through the external circuit by the cathode counterpart to be finally involved in proton reduction into hydrogen.

According to the generalized Marcus theory [6,7,8], the driving force of electron transfer from a donor to an acceptor (in this case, the CB of the semiconductor to the CB of titania) is determined by the difference in their energetic levels. The logarithm of the rate of charge transfer is defined by a quadratic function with respect to the term of charge transfer driving force [4]. By enlarging the energetic difference between these energies, interfacial charge transfer can be boosted.

By aggregating all these requirements, it becomes evident why metal chalcogenides are particularly well-suited for forming semiconductive hetero-nanostructures with TiO_2_ [9], especially when the metal chalcogenide nanostructures consist of nanometal sulfides. Most nanocrystalline metal sulfide compounds exhibit remarkable visible light responsiveness, possess a sufficient number of active sites, and have appropriate reduction and/or oxidation potentials to function as effective photocatalysts [10]. Additionally, their quantum size effects allow for tunable properties such as rapid charge transfer and extended excited-state lifetimes [11].

Among the various combinations with TiO_2_, cadmium sulfide (CdS) nanocrystals have been the most extensively studied [12,13,14,15,16,17,18], despite concerns about their acute toxicity [19,20,21]. Our group, along with several others worldwide, has already prepared CdS-TiO_2_ composites. We demonstrated that replacing pristine TiO_2_ with CdS-TiO_2_ as the photoanode in a standard photoelectrochemical (PEC) cell significantly enhances photocurrent generation, even at 0 V bias [12,22].

In practice, by using a controlled hydrothermal corrosion process followed by air calcination at 400 °C of metallic titanium sheets [1], we successfully produced TiO_2_/Ti substrates. The resulting titania consisted of interconnected nanofibers (NFs), each about tens of nanometers in diameter and several hundreds of nanometers in length, uniformly covering the metal surface. These nanofibers crystallized in the anatase phase, exhibiting strong UV-range absorption. Using metallic Ti plates as both the support and the precursor for TiO_2_ NFs allowed us to streamline the production process while ensuring optimal contact between the photoactive and conductive elements of the photoanode, minimizing current losses [1,23,24,25].

By selecting a heating temperature below 500 °C, we were able to promote the formation of the anatase phase, which is more suitable for our intended application compared to other titania allotropes [26,27]. Additionally, preformed CdS nanoparticles (NPs), approximately 2–3 nm in size, were deposited onto and between the TiO_2_ NFs through a simple ethanol-based impregnation method. This process resulted in a valuable CdS-TiO_2_/Ti composite architecture for the targeted application [12,22]. Impregnation is considered as the simplest and most sustainable approach for constructing semiconductive hetero-nanostructures [28,29,30].

Despite these promising photoelectrochemical (PEC) results, replacing CdS NPs with less toxic yet equally effective metal sulfide NPs nanoparticles remained a key objective of our work. Our goal was to reduce risks to workers, consumers, and the environment during production and handling, ensuring that neither safety nor innovation is compromised, with sustainability playing a central role in our approach.

In this context, we chose to test CuS-TiO_2_/Ti using a similar simple, low-cost material processing method. Copper sulfide compounds, such as Cu_2_S and Cu_7_S_4_, are well known for their unique optical and electrical properties. Specifically, CuS has a narrow band gap with an energy range between 2.0 and 2.2 eV [31,32]. It can exhibit either p-type (predominantly) or n-type conductivity depending on the nature of its self-doping, making it highly promising for heterojunction design. CuS is often non-stoichiometric, meaning that variations in the oxidation state of its components can result in either excess electrons in its CB or holes in its otherwise filled valence band (VB). This property allows nanosized CuS to be classified as a plasmonic semiconductor due to its self-doping characteristics [33,34].

Plasmonic semiconductors, including CuS, possess extraordinary optoelectronic properties, particularly related to localized surface plasmon resonances (LSPRs) in the near-infrared (NIR) spectral region [35,36]. These characteristics make CuS an excellent candidate for photoelectrochemical (PEC) applications, such as light-assisted water splitting, either as a standalone material [37] or when coupled with titania [38].

Such self-doped particles with controlled morphology were previously prepared through wet synthesis routes with variable degrees of success (see, for instance [39,40,41,42,43]). Achieving the desired non-stoichiometry without contamination from foreign phases required several strategies. Typically, the redox properties of the reaction medium were carefully adjusted to attain the appropriate mixed valence states of copper and/or sulfur while avoiding the formation of impurities such as CuO or metallic Cu. Post-synthesis treatments, such as ion exchange or exposure to a redox atmosphere, were also employed to modify the oxidation states of the copper and sulfur elements [38,40,41,42,43,44,45].

Given that one of the primary goals of this study was the straightforward and reproducible production of uniformly sized, well-crystallized, and self-doped CuS NPs, we opted for microwave (MW)-assisted polyol synthesis. By varying the MW heating power and time and using either thioacetamide (TAA) or thiourea (ThU) as sulfur sources, we successfully optimized the synthesis to obtain the desired nanostructures. In this method, Cu^2+^ and S^2−^ precursors were dissolved in a polyol solvent, and rapid heating was applied to the reaction medium. This facilitated nucleophilic substitution and condensation reactions while preventing complete reduction, thereby avoiding the contamination of Cu⁰ [46], and allowing partial reduction of Cu^2+^ to Cu^+^ for the desired covellite CuS self-doping.

Additionally, the adsorption of polyol molecules on the surface of the primary particles inhibited their growth and aggregation, enabling effective size control [47]. Finally, using a simple ethanol-based impregnation process, the CuS NPs were deposited onto and between the TiO_2_ nanofibers (NFs), resulting in the desired CuS-TiO_2_/Ti architectures.

Structural, optical, and electrochemical characterizations of the resulting CuS-TiO_2_/Ti composite confirmed the effectiveness of this material and processing approach. These results contribute to the advancement of photoelectrochemical (PEC) technology, highlighting its potential for sustainable hydrogen production.

## 2. Experiments

### 2.1. Material Synthesis

TiO_2_/Ti sheets were prepared by hydrothermal corrosion of commercial Ti plates. In brief, 0.8 × 2 cm Ti metallic plates (Goodfellow, >97%, 1 mm of thickness) were mechanically polished (intermediate polishing) with sandpaper of different granulometries to ensure the removal of the pre-existing oxide layers and eventual contaminants. The samples were washed in ultrasound in water, ethanol, and acetone (10 min each) and air-dried before being submitted to chemical polishing. In practice, the plates were immersed in an oxalic acid aqueous solution (5% *w*/*w*, equivalent to 0.6 mol·L^−1^) and heated at 100 °C for 2 h. They were then washed in water and air dried before their controlled hydrothermal surface oxidation according to already optimized operating conditions [1]: The plates were placed in a Teflon^®^-lined 120 mL autoclave in an equivolumetric mixture (10 mL total) of H_2_O_2_ (30%) and NaOH (10 mol·L^−1^). The closed autoclave was placed in an oven at 80 °C for 24 h. The plates were rinsed with deionized water, protonated HCL solution (0.1 mol·L^−1^), and again with deionized water and dried at 80 °C. Finally, a calcination in air took place at a 45 min heating ramp, with a target temperature of 400 °C kept constant for 60 min. Scanning electron microscopy (SEM) confirmed the 1D porous network titania morphology covering the entire titanium sheet (between 0.5 and 1.0 µm in thickness), with ropes of an average diameter of 20–50 nm, interweaving between each other, leading to a highly porous hierarchical structure (Appendix A). Transmission electron microscopy (TEM) evidences the veil-type structure on individual titania NFs, each veil being folded on itself, resulting in a fiber structure with a large specific area (Appendix A) and then a large interaction surface, which is an advantage for the desired catalytic application.

CuS particle synthesis was performed by the MW-assisted polyol process under different conditions. In the presence of sulfide nucleophilic agents, microwave heating allows shortening the reaction, promoting sulfidation instead of total reduction [46]. Two sulfide sources were used (ThU and TAA), and two different operating conditions were explored: low heating power, typically 200 W, for relatively long reaction times (25 to 30 min), and high heating power, namely 1200 W, for very short reaction times (1 to 3 min). In practice, 6.15 × 10^−2^ mol·L^−1^ of copper (II) acetate were dispersed in 80 mL of ethyleneglycol (EG) with either TAA or ThU (TAA or ThU/copper molar ratio being equal to 1.2). The mixture was vigorously agitated and submitted to intense ultrasound for at least 30 min. The mixture was then transferred to a microwave-adapted reactor and heated in a multiwave Anton Paar microwave oven under constant radiation power. The resulting particles were recovered by centrifugation and washed with ethanol at least three times. They were finally dried at 60 °C overnight in air. The list of the prepared samples is summarized in the Appendix A, in which each sample is referenced by adding to CuS the type of sulfur source, microwave power, and heating time. For instance, CuS-ThU-1200-1 corresponds to particles prepared with ThU for a heating time of 1 min under a heating power of 1200 W. All the produced particles are from the covellite structure, as confirmed by Rietveld refinements on all the recorded X-ray diffraction (XRD) patterns (Appendix A and Appendix A). The smallest CuS particles were selected for the final photoanode preparation step to take advantage of their large specific surface area. According to TEM micrographs of ethanolic suspensions containing the variously prepared particles (Appendix A), using a low microwave power of 200 W with a long reaction time (25 min) resulted in larger particle sizes (up to ~40 nm). In contrast, a higher power of 1200 W with a shorter reaction time (1 min) significantly reduced the particle size (down to ~7 nm). Additionally, for a given reaction time, particles synthesized with the thiourea (ThU) precursor were consistently smaller than those produced with thioacetamide (TAA), due to the faster decomposition of ThU in the reaction medium [47]. As a result, the CuS-ThU-1200-1 particles, with a typical size of 7–8 nm, were chosen for the fabrication of the CuS-TiO_2_/Ti photoanode.

The previously prepared TiO_2_/Ti substrates were fully immersed in a dilute CuS impregnation solution (3 mg of CuS in 4 mL of ethanol), sonicated for 10 min, and then left to rest overnight. This low concentration was deliberately chosen to allow for a performance comparison between our engineered CuS-TiO_2_/Ti photoanode and a similarly prepared photoanode in which the less toxic CuS co-catalysts were replaced with the more toxic CdS ones [12,22]. After impregnation, the plates were rinsed with ethanol, dried at 80 °C for 1 h, and stored under standard conditions without requiring any special handling.

### 2.2. Material Characterization

The structure of all the prepared samples was examined by XRD using two diffractometers (Panalytical, Almelo, Netherlands), an Empyrean equipped with a Cu Kα X-ray source (1.5418 Å) operating in the w-2θ (w = 1°) geometry for the plates and an X’pert Pro equipped with a Co Kα X-ray source (1.7889 Å) operating in the θ-θ geometry for the powders. The collected patterns were analyzed thanks to Highscore+ software version 5.2.0 (PANAYTICAL^©^, Almelo, The Netherlands).

The chemical composition was investigated by X-ray photoelectron spectroscopy (XPS) on an Escalab250 instrument (Thermo-VG, East Grinstead, UK) equipped with an Al-K_α_X-ray source (1486.6 eV). The pass energy was maintained at 200 eV for the survey scan (step size = 1 eV) and at 80 eV for the high-resolution spectra (step size = 0.1 eV). The spectra were calibrated against the (C-C/C-H) C 1s component set at 285 eV, and their analysis was achieved thanks to Avantage software, version 5.9902 (Thermo Scientific™, Boston, MA, USA).

The exact morphology of CuS NPs and TiO_2_ NFs was checked by TEM using a JEM 2100 Plus microscope (JEOL, Tokyo, Japan) operating at 200 kV. Additionally, SEM was carried out on the as-produced pristine TiO_2_/Ti and composite CuS-TiO_2_/Ti photoanodes, using a Gemini SEM 360 microscope (ZEISS, Jena, Germany) operating at 5 kV to check their general morphology. The microscope is also equipped with an Oxford Instrument (Abingdon, UK) energy-dispersive X-ray spectroscopy (EDX) detector (Ultim Max 170 mm^2^ detector), allowing chemical analysis, including chemical mapping. All The recorded micrographs were analyzed by ImageJ software version 1.54 j (open source).

### 2.3. Photoelectrochemical Assays

Each prepared photoanode, TiO_2_/Ti or CuS-TiO_2_/Ti, was employed as a working electrode (WE) in a home-made quartz single-compartment PEC cell (Figure 1), using an Ag/AgCl reference electrode (RE), a Pt wire counter electrode (CE), and a Na_2_SO_4_ aqueous electrolyte ([SO_4_] = 0.5 M, pH = 7). In practice, the I-V curves, thanks to a AUTOLAB PGSTAT12 scanning potentiostat (Metrohm Instrument, Herisau, Switzerland), were collected. Prior to all experiments, the electrolyte was purged by Argon from dissolved dioxygen. To simulate a solar light exposition, a 150 W Xenon lamp (ORIEL instruments, Bozeman, MO, USA) was used, fixing the area of WE illumination to 0.7 × 1.0 cm^2^.

Prior to PEC measurements, the UV-visible diffuse reflectance spectra of the produced composites were recorded on a Lambda 1050 spectrophotometer (PerkinElmer, Shelton, CT, USA) equipped with a PTFE-coated integration sphere.

## 3. Results and Discussion

### 3.1. Photoanode Engineering

The elaboration of the CuS-TiO_2_/Ti substrates consisted of three main steps involving Ti plate-controlled corrosion to produce a well-adherent thick, porous anatase coating on a conductive substrate, polyol CuS particle synthesis optimization to obtain ultrafine co-catalysts (less than 10 nm in size), and an easy-to-achieve impregnation route, tacking advantage from the abundance of pores and the high surface-to-volume ratio of pristine TiO_2_/Ti.

The efficiency of the photoanode material processing was first checked by XRD analysis (Figure 2). The recorded pattern of CuS-TiO_2_/Ti matched very well with that of pure TiO_2_ and Ti phases. Indeed, all the diffraction peaks were fully indexed in the tetragonal anatase structure (ICDD No. 00-021-1272) and the hexagonal titanium one (ICDD No. 00-044-1294) without clear evidence of CuS signature due to its low content and/or its ultrasmall crystal size.

To confirm the presence of CuS particles, the engineered photoanode was observed by SEM, and the recorded SEM micrographs were compared to those collected on pristine TiO_2_/Ti. A simple contrast lecture of the two types of images evidenced some differences on some titania fiber nodes (Figure 3). Focusing on such a zone, EDS chemical mapping confirmed the simultaneous presence of copper and sulfur elements at this area at almost the same concentration (Figure 4).

A semi-quantitative EDS analysis of CuS-TiO_2_/Ti confirmed the presence of copper and sulfur elements at very low but non-zero contents compared to titanium and oxygen elements (Appendix A), leading to a whole Cu/Ti content of about 0.5 at.-%. Such an atomic ratio aligns well with the low concentration of the CuS impregnation solution used in the photoanode preparation. This ratio is also comparable to the Cd/Ti atomic ratio in our previously studied CdS-TiO_2_/Ti photoanode, making a performance comparison between the two systems in terms of PEC efficiency both relevant and meaningful.

Additionally, a comparison of the Cu/Ti atomic ratio from EDS with that inferred from XPS analysis confirmed that a significant portion of the impregnated CuS particles reside on the outer surface of the TiO_2_ fibers (15.4 at.% vs. 0.5 at.%), which is advantageous for our intended application. The survey XPS spectrum of CuS-TiO_2_/Ti, compared to those of pristine TiO_2_/Ti and CuS (Figure 5a), confirms the presence of all expected elements—Ti and O for the TiO_2_ phase, and Cu and S for the CuS phase. While there were no notable differences in the respective bonding energies between samples, a significant variation in the Cu2p and S2p peak intensities was observed. Specifically, the surface Cu concentration on CuS-TiO_2_/Ti was 2.9 at.%, compared to 23.2 at.% on the surface of pristine CuS.

A focus on the high-resolution Cu 2p signal recorded on CuS-TiO_2_/Ti (Figure 5b) compared to that of pristine CuS (Appendix A) confirms that copper is at the particle surface divalent with Cu 2p_1/2_ and Cu 2p_3/2_ binding energies of 952.5 and 932.4 eV, respectively, close to the values reported in the literature for CuS [48,49] and Cu_2_S [50,51] phases. CuS also exhibits a small shake-up or multiplet splitting structure, while Cu_2_S does not [48,49,50,51,52]. This feature agrees with the formation of CuS without excluding the presence of Cu^+^ species. Additionally, the Cu LMM peaks (Appendix A) recorded on all the prepared CuS particles, including those used for the preparation of CuS-TiO_2_/Ti, exhibit a peak shape completely different from that usually observed on Cu^0^ species [52], confirming the absence of copper metal. Moreover, the slight non-stoichiometry measured by XPS on the CuS particles before and after their attachment by impregnation to the pristine TiO_2_/Ti (Table 1) agrees fairly with self-doping, which may result from a partial substitution of Cu^2+^ cations by monovalent Cu^+^ ones within the covellite lattice.

Assuming that all these features are representative of the whole volume of all CuS particles (the 7–8 nm average size of CuS particles is smaller than the 10–12 nm XPS analysis depth), one may conclude in favor of their self-doping, giving then of the properties of plasmonic semiconductors.

Also, the S2p high-resolution XPS spectra of both CuS (Appendix A) and CuS-TiO_2_/Ti (Figure 5b) are quite similar. Their total intensities are of course different, but both exhibit a doublet at 163.5 (2p_1/2_) and 162.3 eV (2p_3/2_) characteristics of sulfide S^2−^ species in CuS [48,49] or Cu_2_S [50,51] phases. Interestingly, both exhibit a supplementary contribution: a broad and small in intensity peak at 168.6 eV usually attributed to sulfate SO_4_^2−^ anions, suggesting a weak surface oxidation with the production of a thin CuSO_4_ passivation layer. In other words, the composition of the analyzed copper sulfide particles is consistent with a CuS@CuSO_4_ core-shell nanostructure. Comparing the intensity of the S^2−^ 2p_3/2_ and SO_4_^2−^ 2p_3/2_ XPS peaks allows us to estimate that, approximatively, the fourth of the involved sulfur atoms are in the form of sulfate, in agreement with a very thin protective copper sulfate layer.

If the former XPS analysis confirmed the presence of CuS particles on the surface of the CuS-TiO_2_/Ti sample, it also suggested that the chosen impregnation route did not affect the chemical state of titanium cations on the surface of the titania coating. Indeed, there are no significant differences between the Ti 2p XPS profiles of TiO_2_/Ti and CuS-TiO_2_/Ti, as well as between their O 1s XPS profiles (Appendix A), agreeing very well with the TiO_2_ oxide nature of the outer layer of the titanium plates [1,5,53].

Finally, the optical absorption spectrum of the engineered CuS-TiO_2_/Ti photoanode was measured in diffuse reflectance and compared to that of pristine TiO_2_/Ti, recorded in diffuse reflectance as well, and that of pristine CuS, recorded in a transmission scheme (Figure 6). Regarding the semiconducting nature of TiO_2_/Ti, the typical anatase band-to-band signature at around 300 nm was identified, with a band-gap value inferred from Tauc plots of about 3.2 eV in pristine TiO_2_/Ti and 2.6 eV in CuS-TiO_2_/Ti. The last small value is not at all the consequence of a gap decrease but is the consequence of a more complex composite band diagram. Indeed, the CuS-TiO_2_/Ti spectrum is the combination of those of pristine TiO_2_/Ti and CuS, with absorption capabilities ranging from UV to NIR spectral ranges, due to the photo-excitation of both titania and copper sulfite semiconductors. The anatase and the covellite band-to-band absorptions (around 300 [1,5] and 400 nm [30], respectively) superposed to the self-doped CuS LSPR absorption (around 1200 nm [31,32]) explain together the optical properties of our engineered photoanode. Clearly, the amount of CuS particles deposited on TiO_2_/Ti by impregnation, even small, appeared large enough to induce a widened light absorption, which is fruitful for improved PEC responses.

### 3.2. Photoanode PEC Properties

Hydrogen photo-generation activity of the as-synthetized CuS-TiO_2_/Ti photoanode and its TiO_2_/Ti parent was carried out under a Xenon lamp irradiation using a passive and neutral electrolyte. Interestingly, operating in a passive electrolyte, an intermittent illumination of CuS-TiO_2_/Ti provides a higher photocurrent than pristine TiO_2_/Ti, whatever the applied bias (Figure 7). The chronoamperometry under intermittent lighting (black arrows indicating the beginning of dark periods and the orange arrows indicating the beginning of illuminated periods) shows that the sensible increase in the photogenerated current is stable at 1.23 V (vs. Ag/AgCl), and while there is a decrease in the photocurrent during the first minute of illuminated periods, the original values are restored after a dark period.

By comparing the behavior of the bare and impregnated photoanodes under 0 V and 1.23 V bias, we can infer that the improved photocurrent is due to a higher amount of photogenerated charge carriers, favored by the increase of absorbed photons, promoted by CuS NPs. CuS-TiO_2_/Ti absorbs more light, in a wider spectral range, than pristine TiO_2_/Ti, creating a sufficient number of electron−hole pairs. The electrons can then be transferred from CuS CB to that of TiO_2_ as summarized hereafter:CuS + hv → CuS(e^−^ + h^+^)(1)
e^−^(CB_CuS_) + TiO_2_ → TiO_2_(e^−^)(2)
h^+^(VB_TiO_2__) + H_2_O → 2H^+^ + ½O_2_(3)
h^+^(VB_TiO_2__) + CuS → CuS(h^+^)(4)
h^+^(VB_CuS_) + H_2_O → 2H^+^ + ½O_2_(5)

The collected electrons in TiO_2_ CB were then transferred through the external circuit to the Pt cathode to achieve the reduction of aqueous protons into hydrogen gas (Figure 8).

These findings align with results from a few research groups studying CuS-TiO_2_ systems, which remain relatively underexplored in the literature on photocatalytic hydrogen generation compared to metal chalcogenide-based titania nanocomposites, such as the CdS-TiO_2_ system. This is despite the well-documented acute toxicity of (see, for instance, [19,20,21] and the references therein).

To the best of our knowledge, notable results have been reported by Chandra et al. [56], who prepared their composites by a hydrothermal and a solution-based process. Operating by photocatalysis (PC) in a sacrificial Na_2_S (0.25 M)-Na_2_SO_3_ (0.25 M) electrolyte, they succeeded in producing 1262 µmol of H_2_ per hour and per gram of catalyst, more than 10 and 9 times higher than that by pristine TiO_2_ and pristine CuS powders under Xe lamp irradiation, respectively. There are also results reported by Jia et al. [57], who decorated TiO_2_ nanowire arrays grown on conductive substrate by CuS nanoclusters by successive ionic layer adsorption and reaction (SILAR method). Operating by photo-electrocatalysis (PEC) in a passive Na_2_SO_4_ (1.00 M) electrolyte, they demonstrated an increased light absorption and an efficient charge separation leading to an improved photocurrent. They succeeded in obtaining within the same setup a photocurrent density 5 times higher than that of pristine TiO_2_ at a bias of 0.35 V. One may also cite the results of Liu et al. [58], who successfully constructed a CuS/TiO_2_ heterojunction using metal-organic framework (MOF)-derived TiO_2_ as a substrate. They pointed out that CuS/TiO_2_ exhibited excellent bifunctional PC activity without noble metal cocatalysts. They typically evidenced H_2_ production and benzylamine oxidation in a coupled experiment, with a H_2_ evolution activity of the CuS/TiO_2_ 17.1 and 29.5 times higher than that of TiO_2_ and CuS, respectively. Wang et al. [59] also investigated CuS/TiO_2_ photocatalysts, prepared via a high-temperature hydrothermal method, and evaluated their photocatalytic activity. They demonstrated that loading TiO_2_ with 1 wt.-% CuS significantly enhanced its photocatalytic performance for water decomposition to hydrogen in a methanol aqueous solution under Xe lamp irradiation. The CuS/TiO_2_ photocatalysts produced approximately 570 µmol of H_2_ per hour, which is 32 times higher than that produced by pristine TiO_2_.

Clearly, in all these studies and in others (Appendix A), widened light absorption and an efficient charge separation were systematically reported. All converged, placing CuS as one of the most interesting metal chalcogenide titania co-catalysts for water splitting.

To support these scientific advances, we compared the performance of the CuS-TiO_2_/Ti photoanode with that of a similarly prepared CdS-TiO_2_/Ti photoanode [12,22]. The main difference between the two is the size of the particles, with CdS having a particle size of 3 nm. Both photoanodes were tested using the same photoelectrochemical (PEC) setup. Interestingly, the photocurrent measured for the CuS-TiO_2_/Ti photoanode was consistently higher than that for the CdS-TiO_2_/Ti photoanode. This is attributed to the broader light absorption range of CuS (Figure 9), indicating that CuS is a more effective co-catalyst compared to CdS.

## 4. Conclusions

In conclusion, the integration of CuS nanoparticles (NPs) with TiO_2_ nanofibers (NFs) has proven to be a promising approach for achieving efficient photoelectrochemical (PEC) responses in water splitting and hydrogen generation, in good alignment with the relevant literature. The synergistic properties of these nanomaterials enable excellent light absorption, effective charge separation, and efficient electron transport, leading to significant improvements in PEC performance.

By optimizing the microwave-assisted polyol process conditions, self-doped covellite CuS particles with sizes of 7–8 nm, which absorb in the visible and near-infrared (NIR) spectral ranges, were successfully produced without foreign contaminants. These particles were effectively integrated with TiO_2_ NFs supported on a titanium substrate, which was prepared through controlled metal plate corrosion (hydrothermal treatment followed by calcination). The simple ethanol-based impregnation method proved sufficient for creating the CuS-TiO_2_/Ti semiconductive hetero-nanostructures.

A CuS concentration as low as 0.5 at.-% in the composite was sufficient to achieve photocurrents of 0.030 and 0.122 mA/cm^2^ at 0 V and 1.23 V, respectively. In comparison, the photocurrents measured for pristine TiO_2_/Ti under the same PEC conditions were 0.020 and 0.051 mA/cm^2^. Notably, the photocurrent with the CuS co-catalyst was comparable to that obtained with toxic CdS at 0 V and significantly higher at 1.23 V (0.122 vs. 0.082 mA/cm^2^). These results highlight that the selected materials and the employed synthetic approaches offer a novel and effective pathway for developing sustainable hydrogen production systems.

## Figures and Tables

**Figure 1 nanomaterials-14-01581-f001:**
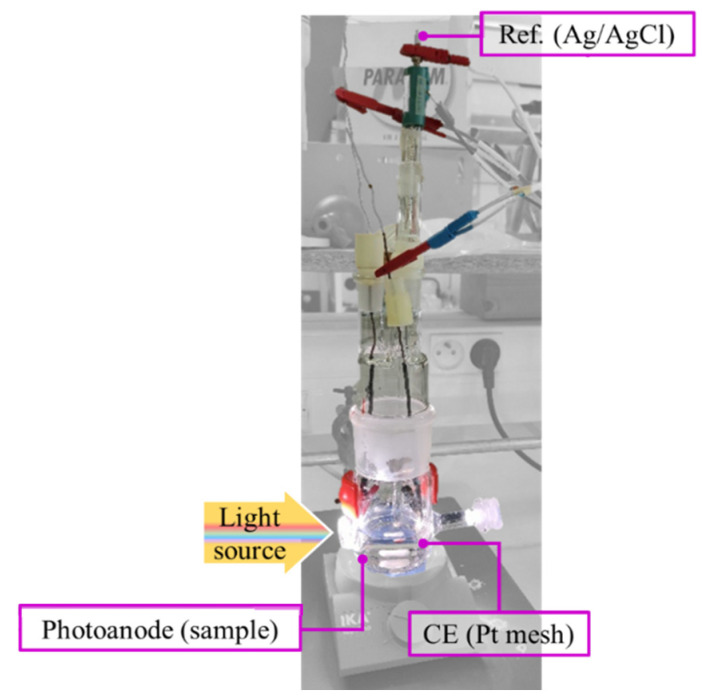
Home-made single-compartment quartz PEC cell, working in a classic 3-electrode configuration, using Ag/AgCl RE and Pt CE.

**Figure 2 nanomaterials-14-01581-f002:**
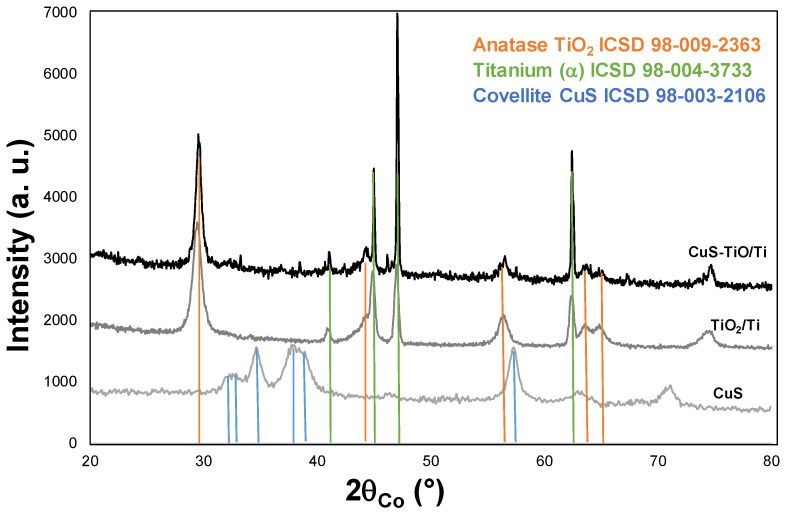
XRD patterns of as-prepared CuS-TiO_2_/Ti, TiO_2_/Ti, and CuS. The peak positions of TiO_2_ (anatase), Ti (α), and CuS (covellite) references are given for information.

**Figure 3 nanomaterials-14-01581-f003:**
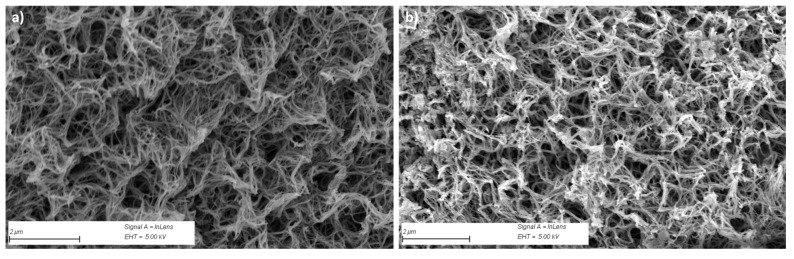
Top view SEM micrographs of TiO_2_/Ti (**a**) before and (**b**) after CuS impregnation, highlighting the presence of an additional contrast at some TiO_2_ fiber nodes.

**Figure 4 nanomaterials-14-01581-f004:**
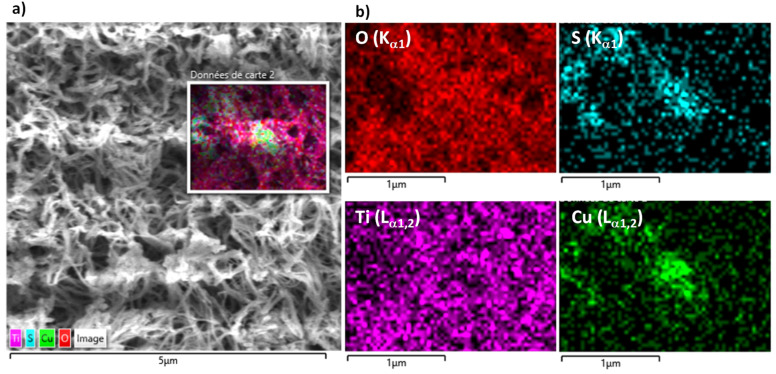
SEM-EDX analysis of the CuS-TiO_2_/Ti sample: (**a**) Z-contrasting top view SEM micrograph highlighting a TiO_2_ fiber noddle on which an assembly of CuS particles is aggregated, (**b**) EDS chemical mapping confirming the copper and sulfur element co-concentration in the selected area, in agreement with the presence of CuS particles.

**Figure 5 nanomaterials-14-01581-f005:**
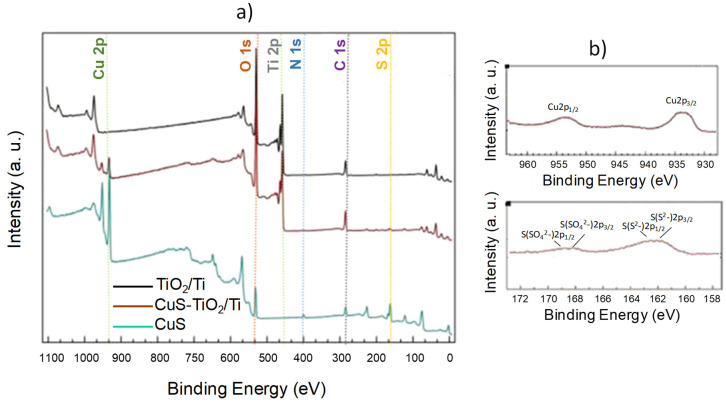
(**a**) Survey XPS spectra of CuS-TiO_2_/Ti (brown line), TiO_2_/Ti (black line), and CuS (blue-green line). (**b**) Cu 2p and S 1s XPS high-resolution spectra of CuS-TiO_2_/Ti (brown line).

**Figure 6 nanomaterials-14-01581-f006:**
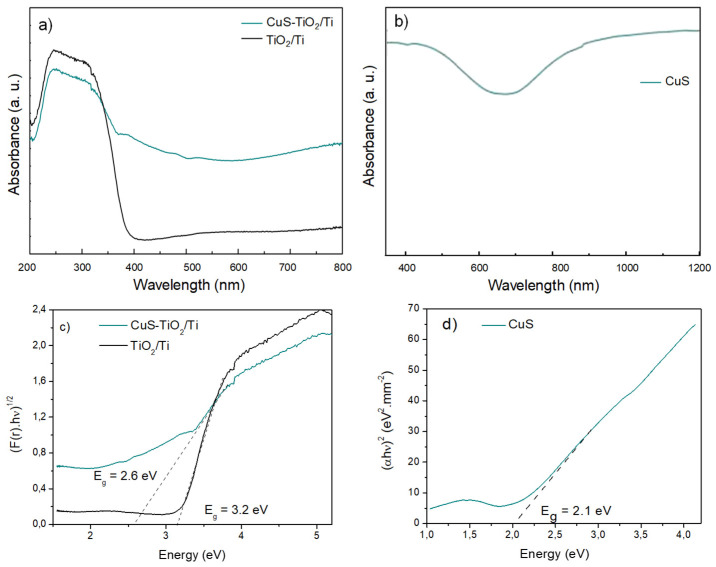
UV-Vis-NIR absorption spectra of (**a**) CuS-TiO_2_/Ti and TiO_2_/Ti recorded in total reflectance mode compared to (**b**) that of pristine CuS recorded in transmission. (**c**,**d**) The Tauc plots inferred from the previous data are given for band-gap determination. The lamp change from UV to visible range during spectra acquisition proceeded at 320 nm.

**Figure 7 nanomaterials-14-01581-f007:**
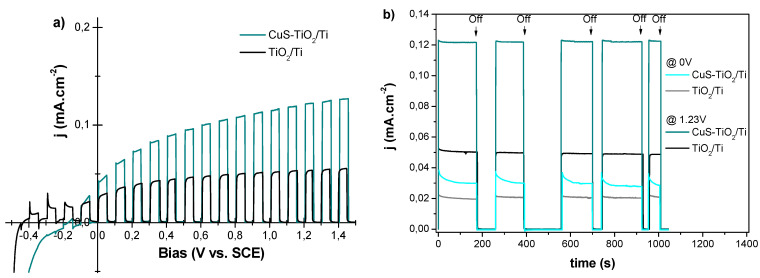
(**a**) Linear sweep voltammetry (10 mV.s^−1^) and (**b**) chronoamperometry of TiO_2_/Ti (black line) are CuS-TiO_2_/Ti (blue-green line) in a passive Na_2_SO_4_ (0.5 M) electrolyte.

**Figure 8 nanomaterials-14-01581-f008:**
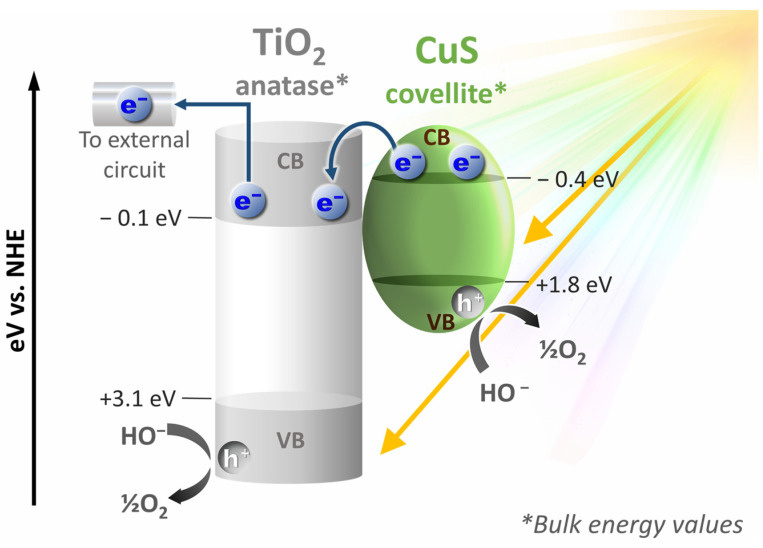
General scheme of the energy band diagram of bulk TiO_2_ anatase and CuS covellite versus the normal hydrogen electrode (NHE), highlighting the reaction of their VB holes with water molecules to produce O_2_ and the collection of their CB electrons for their transfer to the external circuit in a standard PEC cell. To build this diagram, band gap energies and band positions versus NHE of anatase TiO_2_ and covellite CuS were inferred from [54,55], respectively.

**Figure 9 nanomaterials-14-01581-f009:**
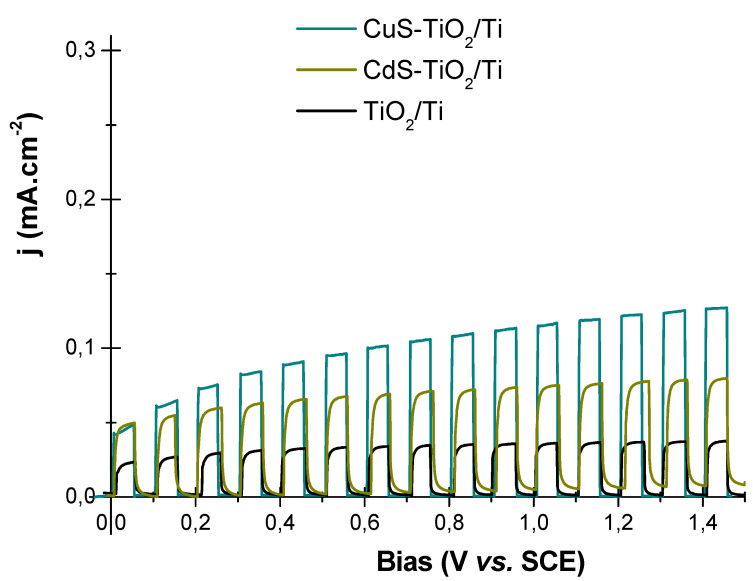
Linear sweep voltammetry (10 mV.s^−1^) of CuS-TiO_2_/Ti (blue-green line), CdS-TiO_2_/Ti (green line), and TiO_2_/Ti (black line) in a passive Na_2_SO_4_ (0.5 M) electrolyte, focusing on the 0 to 1.5 V bias range.

**Table 1 nanomaterials-14-01581-t001:** Recapitulative table of binding energies and atomic compositions for CuS-TiO_2_/Ti photoanodes and their pristine TiO_2_/Ti and CuS counterparts.

	Binding Energy (eV)			Content (at.- %)		
	TiO_2_/Ti	CuS-TiO_2_/Ti	CuS	TiO_2_/Ti	CuS-TiO_2_/Ti	CuS
C 1s (C-C/C-H)	284.8	284.8	284.8	17.6	19.0	13.4
C 1s (C-O)	286.4	286.5	286.5	3.9	4.9	5.9
C 1s (C=O)	288.8	288.5	289.1	2.0	2.0	2.0
Cu LMM	-	565.3	568.9	-	-	-
Cu 2p	-	933.6	932.2	-	2.9	23.2
N 1s	400.1	399.7	399.8	0.4	0.6	4.2
O 1s	529.8	530.0	532.2	53.7	49.5	23.7
S^2−^ 2p	-	162.2	162.5	-	1.7	21.6
SO_4_^2−^ 2p	-	168.5	168.9	-	0.4	5.8
Ti 2p	458.5	458.6	-	22.4	19.0	-

## Data Availability

The raw data supporting the conclusions of this article will be made available by the authors on request.

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
