# Peer review of "Novel Synthesis Route of Plasmonic CuS Quantum Dots as Efficient Co-Catalysts to TiO2/Ti for Light-Assisted Water Splitting"

_nanomaterials, 2024, doi:10.3390/nano14191581_

Round 1

Reviewer 1 Report

Comments and Suggestions for Authors

The authors synthesized self-doped CuS nanoparticles (NPs) via microwave-assisted polyol synthesis to act as co-catalysts with TiO2 nanofibers (NFs) photoanodes, aiming to achieve higher photocurrents for visible light-assisted water electrolysis. The impregnation of the CuS nanoparticles onto TiO2 NFs, which were synthesized through hydrothermal corrosion of a metallic Ti support, resulted in composites with enhanced visible and near-infrared light absorption compared to the pristine support. This modification allows for improved overall efficiency of water oxidation (and consequently hydrogen generation at the Pt counter electrode) in a neutral electrolyte (pH=7) even at 0 V bias. However, some of the results lack in-depth analysis and discussion. I recommend the publication of the paper only after the following issues are appropriately addressed.

1. While the authors provide the morphology of TiO2 NFs, TiO2/Ti sheet, and CuS particles, there is no morphology of the CuS-TiO2/Ti after impregnation. Is the morphology of pristine TiO2 affected after the impregnation process? Additionally, it is recommended to label the lattice stripes of CuS and TiO2 separately in the TEM image.

2. What does the peak in the 550~800 nm region in Figure 5a correspond to?

3. The authors claim that electrons are transferred from CuS CB to TiO2, but this is not supported by sufficient evidence. It is suggested to provide the work functions for CuS and TiO2, as well as their energy band structures, to support this claim.

4. The authors compared the activities of CuS-TiO2/Ti and TiO2/Ti. What is the exact mass ratio of CuS to TiO2/Ti? Have the authors explored the activity differences with varying ratios of CuS-TiO2/Ti? Additionally, what are the exact overall water oxidation efficiencies of each sample?

5. It is recommended to provide a mechanism diagram of light-assisted water splitting to better illustrate the interaction between CuS and TiO2/Ti.

6. Please ensure consistency in the layout and formatting of the images.

Comments on the Quality of English Language

Moderate editing of English language required.

Author Response

Please find enclosed the answers to referee comments file

Reviewer 2 Report

Comments and Suggestions for Authors

This article reports the synthesis of self doped CuS NPs and their electrochemical performance as co catalysts for TiO2/Ti photoanodes. The main point of this article is that the combination of TiO2/Ti NFs and CuS NPs in nanocomposites can improve the UV visible near-infrared absorption region and achieve higher photocurrent in the designed PEC photoanode, thereby enhancing the photocatalytic hydrogen production efficiency. After thorough review, this reviewer did not find that the current work is suitable for publication anywhere in its current form. This may sound harsh, but there are many key factors and obvious errors that can affect their conclusions, which the author did not consider. Therefore, in the opinion of this reviewer, there is still a long way to go for this paper to be considered for publication. The following points are what the reader sincerely hopes the author can revise before the next submission.

1.       The novelty of this work is very low. There are many research groups have reported the photochemical property of CuS /TiO2 nanocomposite catalyst. What is the difference between present work and others?

2.       The article has significant formatting issues. Do authors really pay attentions to write this manuscript?

3.       It was noticed that there are many “similarities” between the synthesis of TiO2/Ti and previous studies published by the same group. To enhance the independence of each article, this reviewer suggests that the authors highlight unique contribution of this study. Is it really necessary to repeatedly describe the synthesis of TiO2/Ti in multiple parts of every article?

4.       When analyzing the UV Vis NIR absorption spectra in the following text, the author proposes the following viewpoint: there is no significant difference between the spectra of CuS and CuS-TiO2/Ti. Which part of the spectrum do authors want to express, from 400nm to 1200nm? Based on the two UV Vis NIR absorption spectra provided by the author, this reviewer does not see that the absorption spectra of CuS and CuS-TiO2/Ti is similar, especially in the range between 600-800 nm range.

5.       The author mentioned that in Figure 7 (b) of this article, black and orange arrows are used to represent the beginning of the dark cycle and the beginning of the illumination cycle, respectively. Specifically, the author conducted chronoamperometry tests on CuS-TiO2/Ti and TiO2/Ti under bias voltages of 0V and 1.23V, respectively, and attempted to use these two arrows to represent the periods of different light sources in the four curves. When testing the dark and lighting cycles of different materials, the authors did not maintain that the cycles of different materials within the same timeframe. The measurement (light on and off) seems random and did have scientific reason behind. In order to improve the accuracy and readability of the article, please ensure the rational design of experiment as well as the accurate analysis.

Comments on the Quality of English Language

average

Author Response

(The authors gave the same response as above.)

Reviewer 3 Report

Comments and Suggestions for Authors

In this manuscript, Chaperman et al. present a novel approach to synthesizing self-doped CuS nanoparticles (NPs) using a microwave-assisted polyol synthesis method aimed at enhancing the performance of TiO2 nanofiber (NF) photoanodes for visible-light-driven water electrolysis. The authors address the copper reduction and nanoparticle size control through a strategic copper cation sulfidation process, resulting in composites claiming significantly improved light absorption and water oxidation efficiency, even under neutral pH conditions and without applied bias. Before considering acceptance, some issues should be addressed:

1. The introduction is very limited. To provide better context and comparison, I suggest the authors cite relevant literature, particularly manuscripts discussing metal sulfides and their combination with semiconductors. https://doi.org/10.3390/catal12111316

2. While the characterization section compares the effects of different sulfur sources, the PEC results do not reflect this comparison. Including this would strengthen the manuscript.

3. Figure 7b is not easily understandable; it would be helpful if the authors could replot it for clarity.

4. A Tauc plot would be beneficial to understand the band gap changes in the single and hybrid semiconductors presented.

5. In Figure 8, it is unclear whether the results are from literature or the author's own work. If from the literature, proper citations with copyright acknowledgments are required.

6. I suggest the authors include XRD results for the CuS synthesized using different methods in the manuscript to illustrate the structural differences better.

7. The naming of the samples is confusing, as it is unclear in the results which CuS nanoparticles were chosen given the two sources used. Clarification is needed.

8. The conclusion is too brief and does not adequately cover all the findings of the study. Expanding this section would improve the manuscript.

Comments on the Quality of English Language

acceptable 

Author Response

Please find enclosed the answers to refreree comments file

Reviewer 4 Report

Comments and Suggestions for Authors

This work reported the synthesis of plasmonic CuS quantum dots as co-catalysts of TiO2/Ti for photoelectrochemical applications. The topic of the work was significant with regard to the development of photoelectrochemical technology. The manuscript can be considered for publication after the following issues are addressed.

Comments:

(1) The irradiation power used for photoelectrochemical reactions should be measured and provided.

(2) Was there an optimal CuS content for maximizing the photoelectrochemical activity of TiO2/Ti?. The authors may consider the effectiveness of interfacial charge transfer by the quantitative effect. For quantitative effect, please refer to reference: Coordination Chemistry Reviews, 2021, vol.438, pp.213876.

(3) The long-term stability of CuS-TiO2/Ti in the photoelectrochemical cell should be examined. The crystallographic structure and the chemical states of the sample upon repeated usage should also be examined with XRD and XPS.

(5) A band alignment illustrating the charge transfer mechanism for CuS-TiO2/Ti should be proposed. With this illustration, it would be easier to understand the origin behind the enhanced photoelectrochemical activity.

(6) The solar-to-H2 conversion efficiency (e.g. IPCE or ABPE) should be measured to provide a more reliable index of photoelectrochemical activity. A table summarizing the performance comparison among the state-of-the-art TiO2-based and CuS-based photoelectrodes should also be provided to highlight the merits of the current work.

(7) CuS and its derivatives, such as Cu2S and Cu2-xS, have peculiar optical properties resembling surface plasmon resonance that can be exploited to enhance photocatalytic activity. Recent development of the practical use of dual-plasmonic Au@Cu7S4 photocatalysts can be introduced to enlighten the readers.

(8) Recent review articles summarizing the current challenge and future perspectives of photoelectrochemical technology could be briefly introduced. For example, topics on “semiconductor nanoheterostructures for photoconversion applications”, “density-functional theory studies on photocatalysis and photoelectrocatalysis”, and “photocatalytic and photoelectrochemical glycerol oxidation towards valuable chemical products” can be highlighted.

Author Response

(The authors gave the same response as above.)

Round 2

Reviewer 1 Report

Comments and Suggestions for Authors

The authors have carefully responded to my concerns and the revised manuscript displays great improvement. Thus, it can be recommended for publication in this journal.

Comments on the Quality of English Language

The English writing basically meets the requirements for paper publication and can be appropriately refined further. 

Reviewer 2 Report

Comments and Suggestions for Authors

The authors made significant efforts to revise this manuscript. It can be published in current form.

Comments on the Quality of English Language

fine.

Reviewer 3 Report

Comments and Suggestions for Authors

Authors revised the manuscript adequately. I suggest its acceptance. 

Reviewer 4 Report

Comments and Suggestions for Authors

The revised manuscript is now in a good shape for publication.